# A study of the effects of job insecurity on organizational citizenship behavior based on the chained mediating effects of emotional exhaustion and organizational identification

Jing Zhu[1]*, Mingfa Yang[2]

**1** Teaching Quality Assessment Center, Chengdu Sport University, Chengdu, China, **2** School of Physical Education, Chengdu Sport University, Chengdu, China

* jessie@cdsu.edu.cn

## Abstract

### Objectives

This study aimed to explore the influence mechanism of job insecurity on organizational citizenship behavior (OCB). Specifically, it sought to examine the chained mediating role of emotional exhaustion and organizational identification in this relationship.

### Methods

A longitudinal time-lagged survey was conducted on 330 employees at two time points. The data were collected using established scales for job insecurity, emotional exhaustion, organizational identification, and OCB. Structural equation modeling (SEM) was used to test the hypothesized chained mediation model.

### Results

The results confirmed that emotional exhaustion and organizational identification are crucial mediators. Three significant indirect pathways were identified: (1) a simple mediation path through emotional exhaustion; (2) a simple mediation path through organizational identification; and (3) a chained mediation path where job insecurity increased emotional exhaustion, which in turn decreased organizational identification, ultimately leading to lower OCB. Notably, emotional exhaustion emerged as the most dominant mediating mechanism.

### Conclusions

This study reveals a complex mechanism through which job insecurity impacts OCB, highlighting a sequential process from emotional strain to cognitive detachment. These findings offer important theoretical insights for job stress models and provide

**Data availability statement:** The original data of this study has been uploaded to the Figshare platform under the title "A Study of the Effects of Job Insecurity on Organizational Citizenship Behavior Based on the Chained Mediating Effects of Emotional Exhaustion and Organizational Identification" and can be accessed via https://figshare.com/s/9785664550acc6c076c5.

**Funding:** The author(s) received no specific funding for this work.

**Competing interests:** The authors declare that they have no competing interests.

practical guidance for organizations to mitigate the negative effects of job insecurity by addressing both employee well-being and their sense of belonging.

## 1. Introduction

In an era of increasing global economic uncertainty and rapid organizational change, job insecurity has emerged as a pervasive and critical issue for the modern workforce [1,2]. Defined as an employee's subjective perception of threat regarding the continuity of their job, this organizational stressor has profound negative consequences for both individuals and organizations [3,4]. For instance, recent data from China indicates a significant decline in employment security, with over 40% of professionals expressing deep concerns about their job stability (China Employment Market Prosperity Report, 2023). This prevailing sense of insecurity is particularly detrimental to Organizational Citizenship Behavior (OCB)—discretionary actions that, while not formally rewarded, are crucial for fostering organizational effectiveness, productivity, and a collaborative atmosphere [5,6]. Given the established importance of OCB for sustainable organizational success, understanding the factors that inhibit it is of paramount importance. Recent empirical evidence has further underscored the critical role of OCB across diverse organizational contexts, demonstrating its enhancement through servant leadership and psychological empowerment [7], its significance in challenging institutional environments [8], and its complex relationship with individual characteristics requiring mediational pathways [9].

To effectively address the adverse impacts of job insecurity, researchers must move beyond simply documenting its negative correlation with OCB and delve into the underlying psychological mechanisms that explain how and why this effect occurs. While previous studies have identified individual mediators, such as emotional exhaustion or organizational identification, they have often examined these pathways in isolation [10,11]. This approach overlooks the complex, sequential nature of employees' psychological responses to job-related stress.

Consequently, a significant research gap exists in understanding the chained, or sequential, mediating process through which job insecurity translates into reduced OCB. Prior academic works have tended to avoid exploring these multi-step mediational models, often due to methodological complexity or a lack of an integrated theoretical framework [12]. Specifically, the literature has not adequately investigated whether job insecurity first triggers an affective reaction (i.e., emotional exhaustion), which in turn degrades a cognitive-evaluative state (i.e., organizational identification), ultimately leading to a withdrawal of discretionary behaviors like OCB. This study directly critiques this oversight by proposing and testing such a sequential pathway.

Contrasting with past research that predominantly focused on single-mediator models, this study integrates two prominent theories—Conservation of Resources (COR) Theory and Social Identity Theory—to build a more comprehensive model. We posit that the full story cannot be told without considering the sequential interplay between emotional depletion and social identification. Therefore, the primary objectives of this study are: (1) To confirm the negative relationship between job insecurity

and OCB; (2) To investigate the mediating roles of emotional exhaustion and organizational identification individually; and (3) To examine the novel chained mediating effect of emotional exhaustion and organizational identification in the relationship between job insecurity and OCB (Fig 1).

This paper contributes to academic literature by introducing and empirically validating a chained mediation model that offers a more nuanced explanation of the job insecurity-OCB link. By connecting the affective consequences (emotional exhaustion) of resource loss to the cognitive-evaluative changes in self-concept (organizational identification), our research provides a deeper, more holistic understanding of how organizational stressors unfold to impact employee behavior.

## 2. Theoretical framework and hypotheses development

### 2.1 Job insecurity and organizational citizenship behavior

Job insecurity, conceptualized as a subjective organizational stressor, is characterized by employees' perceived vulnerability and experience of powerlessness regarding potential job instability [1]. Grounded in Conservation of Resources Theory (COR), individuals confronting resource threats implement defensive strategies to mitigate potential loss [10]. Job insecurity represents a significant resource risk, compelling employees to prioritize personal resource preservation over organizational interests, consequently reducing discretionary work contributions, particularly organizational citizenship behaviors beyond formal job requirements [3].

Social Exchange Theory provides additional theoretical insights: Organizational citizenship behaviors fundamentally rely on reciprocal expectations. Job insecurity undermines the organizational psychological contract, causing employees to perceive the organization's failure to fulfill job security commitments, thus diminishing their willingness to reciprocate organizational investments [13].

Empirical evidence consistently substantiates these theoretical propositions. Multiple meta-analyses have documented a robust inverse relationship between job insecurity and organizational citizenship behavior [14]. Longitudinal research demonstrates the cumulative nature of this effect, revealing progressively diminished organizational citizenship behaviors as job insecurity persists [15]. Synthesizing theoretical analysis and extant research evidence, this study posits a significant negative correlation between job insecurity and organizational citizenship behavior. Consequently, we propose:

***Hypothesis 1: Job insecurity is negatively related to organizational citizenship behavior.***

### 2.2 The mediating role of emotional exhaustion

Emotional exhaustion, a critical dimension of occupational burnout, manifests as the depletion of an individual's emotional resources, psychological energy, and professional enthusiasm [16]. The link between job insecurity and emotional

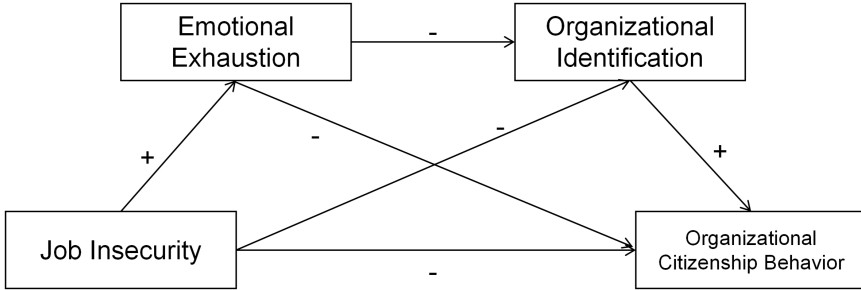

**Fig 1. Hypothetical relationship diagram.**

exhaustion is well-explained by COR theory. Job insecurity represents a persistent threat, compelling employees to persistently allocate cognitive and emotional resources to manage uncertainty, accelerating psychological energy depletion [17]. When resource consumption systematically exceeds resource acquisition, employees enter a resource-loss spiral ultimately culminating in emotional exhaustion [18]. The Job Demands-Resources Model further supports this, conceptualizing job insecurity as a detrimental job demand that weakens psychological resilience by activating energy depletion mechanisms [19]. search has substantiated this theoretical propositi Empirical ren, demonstrating a significant positive correlation between job insecurity and emotional exhaustion [3].

Once emotionally depleted, employees predominantly implement resource-preservation strategies, prioritizing core job responsibilities while simultaneously reducing discretionary organizational citizenship behaviors [20].Emotional resource deprivation fundamentally constrains employees' capacity to focus on collective organizational needs, consequently diminishing discretionary behaviors such as colleague support and organizational development initiatives [21]. Multiple studies have consistently documented the significant negative relationship between emotional exhaustion and organizational citizenship behaviors, a finding corroborated across diverse organizational contexts [22,23].Therefore, we posit that emotional exhaustion serves as a key mechanism transmitting the negative effect of job insecurity to OCB. Consequently, we propose:

***Hypothesis 2: Emotional exhaustion mediates the relationship between job insecurity and organizational citizenship behavior.***

### 2.3 The mediating role of organizational identification

Organizational identification, conceptualized within the framework of Social Identity Theory, represents the degree to which individuals incorporate their self-conceptualization with organizational membership [11]. Social Identity Theory posits that individuals inherently identify with organizations fulfilling their needs for self-perpetuation, self-improvement, and self-differentiation [24]. Job insecurity fundamentally breaches the psychological contract, signaling that the organization cannot provide anticipated security. This erodes the organization's perceived value and its attractiveness as a source of identity, leading employees to psychologically distance themselves from the organization [25].

Moreover, job insecurity can signal potential marginalization, positioning employees as "outsiders" rather than core members, which weakens the foundational basis for organizational identification [26]. The uncertainty and perceived threat induced by job insecurity compel employees to redirect cognitive resources toward personal preservation, further attenuating their psychological connection with the organization [27]. Conversely, organizational identification functions as a critical psychological mechanism connecting individuals to organizational objectives.

Employees with robust organizational identity conceptualize the organization as an extension of their self-definition, perceiving organizational success as synonymous with personal achievement. Consequently, they demonstrate heightened propensity to engage in discretionary organizational citizenship behaviors that benefit the collective [28,29]. Extensive empirical research has consistently validated organizational identification as a significant predictor of OCB [30]. Considering job insecurity's inhibitory effect on organizational identity and the latter's facilitating influence on OCB, organizational identity is expected to function as a mediator.. Consequently, we propose:

***Hypothesis 3: Organizational identification mediates the relationship between job insecurity and organizational citizenship behavior.***

### 2.4 The chained mediating effect of emotional exhaustion and organizational identification

Building upon the preceding arguments, we propose a more complex mechanism involving a sequential mediation pathway. We integrate COR theory and Social Identity Theory to theorize this chain effect. From a COR perspective, job

 

insecurity acts as a chronic stressor that depletes employees' valuable emotional resources, leading to emotional exhaustion [18]. This aligns with Affective Events Theory, which suggests that negative workplace events (job insecurity) trigger negative emotional responses (emotional exhaustion) [31].

The critical link in our proposed chain is the path from emotional exhaustion to organizational identification. Emotional resource depletion constrains the cognitive capacity required to process social information and maintain a positive connection with the organization [27]. When employees are exhausted, their psychological focus narrows from the collective (the organization) to the self (personal survival), making it difficult to maintain a strong sense of oneness with the organization [28]. Empirical research supports this, demonstrating a significant negative relationship between emotional exhaustion and organizational identification [32].

Therefore, the proposed chain-mediated process follows a theoretically substantiated trajectory: job insecurity (a resource threat) leads to emotional exhaustion (resource loss), which in turn weakens organizational identification (eroded social-psychological connection), ultimately resulting in a withdrawal of OCB (reduced discretionary contribution). This mechanism reflects the intricate dynamics of job insecurity's influence, integrating both affective and cognitive pathways. Consequently, we propose:

***Hypothesis 4: Emotional exhaustion and organizational identification sequentially mediate the relationship between job insecurity and organizational citizenship behavior.***

## 3. Methodology

### 3.1 Participants and procedure

The study employed a convenience sampling approach, collecting 330 valid survey responses. The sample comprised 55.15% male participants and 44.85% female participants, demonstrating a relatively balanced gender representation. Age distribution was predominantly characterized by the 26–35 year old cohort (43.94%), complemented by participants under 25 years (26.36%) and 36–45 years (18.48%), representing a comprehensive age spectrum. The educational attainment profile revealed a concentration of middle to higher education backgrounds, with bachelor's degree holders constituting 43.03% and specialist degree holders accounting for 28.48%. Professional experience distribution appeared relatively uniform, with 1–3 years, 4–6 years, and 7–10 years of experience representing 26.06%, 22.12%, and 22.12% of respondents, respectively. Organizational characteristics demonstrated significant diversity. Participant roles were primarily composed of entry-level personnel (57.88%) and junior management (26.67%). Industrial sector representation encompassed key domains including IT/Internet (24.24%), Manufacturing (20.91%), service industry (16.97%), and other sectors. Organization size distribution ranged from large enterprises with over 1,000 employees (25.15%) to small and medium-sized organizations, with 75.15% of respondents being full-time employees.

This study employed a time-lagged design approved by the Ethics Committee of Chengdu Institute of Physical Education (Approval No. 202584). The recruitment period for this study ran from March 31, 2025, to April 10, 2025. All participants provided written informed consent before participating in the study. No minors were involved in this research. Data were collected over a five-week period, comprising three measurement phases: T1 (week 1) assessed job insecurity and demographic attributes; T2 (week 3) evaluated emotional exhaustion and organizational identity; and T3 (week 5) measured organizational citizenship behavior. The research team acquired samples through diverse channels, distributing questionnaires in collaboration with the human resources departments of 12 medium-sized enterprises and utilizing a professional research platform (Questionnaire Star) to enhance the sample reach. To guarantee the precision of data linkage across the three time points, participants were instructed to create and use the same unique anonymous identification code (e.g., first letter of name – last four digits of cell phone number) for each survey. The research team subsequently used these codes to manually match the responses. The questionnaire was electronically delivered via email, allowing 48 hours for completion; those who did not complete the questionnaire within this timeframe were sent a reminder email. To

ensure data quality, the questionnaire included attention-check questions and we screened for invalid data (e.g., excessively short completion times or logical inconsistencies), resulting in 330 complete matches across the three time points. (Note: This study includes other categorical variables as control variables in the model analysis. The comparison results (see appendix) indicate that the parameter estimates with control variables do not differ significantly from those of the baseline model. Considering data consistency and result clarity, this paper reports the estimation results from the baseline model.)

### 3.2 Measures

**3.2.1 Job insecurity scale.** The study employed the validated Job Insecurity Scale [33] to assess employees' subjective perceptions of job stability. The measurement instrument comprises five carefully constructed question items, assessed via a 6-point Likert-type response format (1 = never, 6 = always). The scale comprehensively integrates employees' cognitive appraisals and affective responses, systematically capturing critical dimensions of job insecurity: concerns about job stability, anticipation of potential organizational changes, and apprehension regarding potential dismissal. In the current research context, the scale demonstrated exceptional internal consistency, with a Cronbach's alpha coefficient of 0.916.

**3.2.2 Emotional exhaustion scale.** The study employed the validated Emotional Exhaustion Scale [34] to evaluate employees' psychological resource depletion. The measurement instrument comprises three carefully constructed question items, assessed via a 7-point Likert-type response format (1 = strongly disagree, 7 = strongly agree). The scale systematically assesses the multidimensional aspects of emotional depletion, mental exhaustion, and workplace-induced fatigue that employees encounter during professional engagement. In the current research context, the scale demonstrated robust psychometric properties, with a Cronbach's alpha coefficient of 0.842.

**3.2.3 Organizational identification scale.** The study employed the validated Organizational Identification Scale [35] to assess employees' psychological affiliation and organizational embeddedness. The measurement instrument comprises six carefully constructed items, assessed via a 5-point Likert-type response format (1 = strongly disagree, 5 = strongly agree). The scale comprehensively examines core manifestations of organizational identification, including individuals' affective responses in response to organizational achievements or critiques, perceptions of shared organizational accomplishments, and linguistic markers of organizational belonging (e.g., using "we" instead of "they"). In the original study, the internal consistency reliability was .87 across a sample of 297 alumni, demonstrating robust psychometric properties. In the current research context, the Organizational Identification Scale exhibited an exceptional Cronbach's alpha coefficient of 0.884.

**3.2.4 Organizational citizenship behavior scale.** The study employed the validated Organizational Citizenship Behavior Scale developed [36], originally revised from Podsakoff et al. [37], to assess employees' discretionary workplace behaviors. The comprehensive measurement instrument consists of ten items, assessed via a 7-point Likert-type response format (1 = strongly disagree, 7 = strongly agree), encompassing two critical dimensions: prosocial organizational behaviors (items 1–7) and organizational citizenship ethics (items 8–10). The scale has demonstrated robust cross-cultural validity, with internal consistency coefficients of 0.81 and 0.73 for prosocial behaviors and citizenship ethics subscales, respectively, in a comparative study between Chinese and U.S. contexts. In the current research context, the Organizational Citizenship Behavior Scale exhibited exceptional psychometric properties, with a Cronbach's alpha coefficient of 0.952 (Table 1).

### 3.3 Data analysis

The data processing and analysis of this study employed a multi-stage analytical strategy. The raw data were organized and merged using Microsoft Excel, with precise matching achieved through the participant-provided identification codes. Subsequently, the data were imported into SPSS 26.0 for comprehensive analysis, employing Pearson correlation

**Table 1. Research instrument validation.**

|  | Cronbach's α | KMO value | χ²/df | GFI | RMSEA | RMR | CFI | NFI |
|---|---|---|---|---|---|---|---|---|
| Job insecurity | 0.916 | 0.898 | 1.089 | 0.994 | 0.016 | 0.031 | 1 | 0.995 |
| Emotional Exhaustion | 0.843 | 0.726 | – | 1 | 0 | 0.003 | 1 | 1 |
| Organizational Iidentification | 0.884 | 0.908 | 0.639 | 0.994 | 0 | 0.022 | 1.004 | 0.994 |
| Organizational Citizenship Behavior | 0.952 | 0.965 | 1.68 | 0.967 | 0.045 | 0.078 | 0.991 | 0.978 |

coefficients to assess the bivariate relationships among the study variables and descriptive statistical analysis to delineate sample characteristics and variable distributions. The mediation analysis was grounded in the framework articulated by Hayes (2013), utilizing the PROCESS macro for SPSS (Model 6) to perform a chained mediation analysis and implementing the Bootstrap method (5,000 repetitions, 95% confidence interval) to verify the significance of indirect effects. This method offers robust statistical support for examining how job insecurity influences organizational citizenship behaviors via emotional exhaustion and organizational identity.

## 4. Findings

### 4.1 Common method bias test

To proactively mitigate common method bias (CMB), a three-wave, time-lagged research design was employed, which is a key procedural remedy. We further conducted statistical analyses to assess potential remaining bias. Harman's single-factor test revealed that the first unrotated factor accounted for 49.59% of the variance, falling below the 50% threshold. Additionally, a full collinearity assessment was performed, yielding a maximum variance inflation factor (VIF) of 3.36. While we acknowledge that this VIF value is slightly above the strict 3.3 benchmark suggested by Kock (2015), it remains well within other commonly accepted thresholds (e.g., 5 or 10). Considering our robust time-lagged design as the primary control, coupled with these statistical results, we conclude that common method bias is not a significant concern for this study.

### 4.2 Descriptive statistics and correlation analysis

To investigate the distributional characteristics of the research variables and their interrelationships, descriptive statistics and correlation analysis were conducted on the primary variables in this study. The results indicated that participants reported a moderately high level of job insecurity (M = 3.58, SD = 1.492), exhibited a high degree of emotional exhaustion (M = 3.888, SD = 1.735), showed a moderate level of organizational identification (M = 3.023, SD = 1.056), and attained a high level of organizational citizenship behavior (M = 3.868, SD = 1.696). Pearson's correlation analysis revealed that job insecurity exhibited a significant positive correlation with emotional exhaustion (r = 0.627, p < 0.01) and demonstrated a significant negative correlation with organizational identity (r = −0.550, p < 0.01) and organizational citizenship behavior (r = −0.579, p < 0.01). Emotional exhaustion significantly negatively correlated with organizational identity (r = −0.569, p < 0.01), while organizational identity was significantly positively correlated with organizational citizenship behavior (r = 0.548, p < 0.01) (Table 2). These correlations preliminarily support the proposed chain-mediated theoretical model in which job insecurity influences organizational citizenship behavior through emotional exhaustion and organizational identification.

### 4.3 Chain mediation effect test

In this study, the data were preprocessed and assessed for covariance before testing mediation effects, and the results indicated that the variance inflation factor for each predictor variable was less than 5, suggesting that the data did not exhibit significant multicollinearity issues and were appropriate for examining complex mediation effects. The study

**Table 2. Pearson correlation – standard format.**

| | M | SD | Job Insecurity | Emotional Exhaustion | Organizational Identification | Organizational Citizenship Behavior |
|---|---|---|---|---|---|---|
| Job Insecurity | 3.58 | 1.492 | 1 | | | |
| Emotional Exhaustion | 3.888 | 1.735 | 0.627** | 1 | | |
| Organizational Identification | 3.023 | 1.056 | −0.550** | −0.569** | 1 | |
| Organizational Citizenship Behavior | 3.868 | 1.696 | −0.579** | −0.662** | 0.548** | 1 |

\* p<0.05 \*\* p<0.01.

employed the Bootstrap method with 5000 bootstrap samples to test the direct and indirect effects of job insecurity on organizational citizenship behavior through emotional exhaustion and organizational identity. The results of regression analysis indicated that job insecurity exerted a significant positive predictive effect on emotional exhaustion (β=0.627, p<0.01) and a significant negative predictive effect on both organizational identity (β=−0.319, p<0.01) and organizational citizenship behaviors (β=−0.579, p<0.01). Furthermore, emotional exhaustion significantly negatively impacted organizational identification (β=−0.369, p<0.01) and also had a significant negative relationship with organizational citizenship behavior (β=−0.421, p<0.01). In contrast, organizational identity exhibited a significant positive predictive effect on organizational citizenship behavior (β=0.193, p<0.01) (Table 3). The regression models demonstrated good fit, with R² values of 0.393, 0.385, 0.336, and 0.506, along with adjusted R² values of 0.391, 0.381, 0.334, and 0.501, and F-tests achieving statistical significance (p<0.001) (Fig 2).

**Table 3. Mediation effects model test (n=330).**

| | Emotional Exhaustion | | Organizational Identification | | Organizational Citizenship Behavior | | Organizational Citizenship Behavior | |
|---|---|---|---|---|---|---|---|---|
| | B | t | B | t | B | t | B | t |
| Job Insecurity | 0.627** | 14.579 | −0.319** | −5.726 | −0.579** | −12.874 | −0.209** | −3.989 |
| Emotional Exhaustion | | | −0.369** | −6.624 | | | −0.421** | −7.902 |
| Organizational Identification | | | | | | | 0.193** | 3.896 |
| R 2 | 0.393 | | 0.385 | | 0.336 | | 0.506 | |
| Adjustment R 2 | 0.391 | | 0.381 | | 0.334 | | 0.501 | |
| F-value | 212.54 | | 102.353 | | 165.745 | | 111.137 | |

\* p<0.05 \*\* p<0.01.

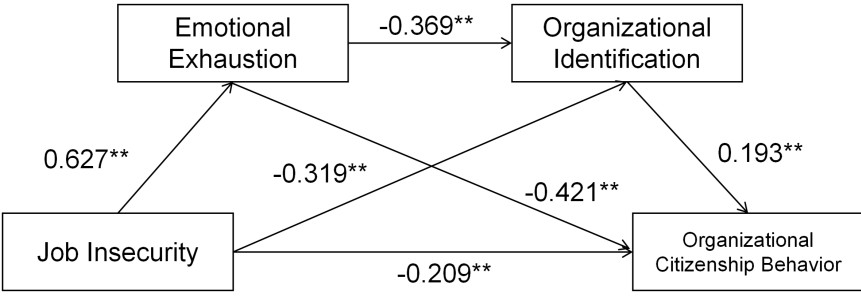

**Fig 2. Path diagram of model.**

The mediation effect analysis revealed that the aggregate effect of job insecurity on organizational citizenship behavior was −0.579, with a direct effect of −0.209 (95% CI = [−0.312, −0.106]) and a total indirect effect of −0.370 (95% CI = [−0.476, −0.265]), which suggests that the mediation variable played a partial mediating role in this influence mechanism. Further analysis indicated that job insecurity exerts an indirect influence on organizational citizenship behavior through three pathways: (1) job insecurity→emotional exhaustion→organizational citizenship behavior, with an indirect effect value of −0.264 (95% CI = [−0.358, −0.170]); (2) job insecurity→organizational identification→organizational citizenship behavior, with an indirect effect value of −0.062 (95% CI = [−0.128, −0.165]); and (3) job insecurity→organizational identification→organizational citizenship behavior, with an indirect effect value of −0.062 (95% CI = [−0.128, −0.168]). The 95% confidence intervals for all three indirect pathways did not contain 0, indicating that these mediating effects were statistically significant (Table 4).

Comparative analyses of the three mediating pathways (C1, C2, C3) further revealed the relative importance of the different mediating mechanisms. The results showed that the single mediation effect of emotional exhaustion was substantially greater than that of organizational identification (C1 = −0.202, 95% CI = [−0.323, −0.078]) and significantly larger than the chain mediation effect (C2 = −0.219, 95% CI = [−0.335, −0.092]); however, the difference between the single mediation effect of organizational identification and the chain mediation effect did not achieve statistical significance (C3 = −0.017, 95% CI = [−0.074, 0.032]). This implies that during the process whereby job insecurity impacts organizational citizenship behavior, the mediating role of emotional exhaustion is the most prominent, explaining 45.60% of the total effect, while the chain mediation effect, although smaller, remains statistically significant, substantiating that job insecurity primarily induces emotional exhaustion in employees, which subsequently reduces organizational identification and ultimately diminishes organizational citizenship behavior. This finding enhances the comprehension of the underlying mechanism through which job insecurity affects organizational behavior and illustrates the pathways of cognitive-emotional linkage.

## 5. Discussion

Grounded in Conservation of Resources (COR) theory and Affective Events Theory (AET), this study explored the intricate mechanism of job insecurity's influence on organizational citizenship behavior (OCB). Via a comprehensive survey of 330 corporate employees and employing rigorous Bootstrap methodology, the research revealed that job insecurity not only

**Table 4. Results of parallel mediation effect test.**

| Effect path | Effect | Boot SE | Boot LLCI | Boot ULCI |
|---|---|---|---|---|
| Total effect | −0.579 | 0.045 | −0.668 | −0.491 |
| Direct effect | −0.209 | 0.052 | −0.312 | −0.106 |
| Total indirect effect | −0.37 | 0.054 | −0.476 | −0.265 |
| Ind1 | −0.264 | 0.048 | −0.358 | −0.17 |
| Ind2 | −0.062 | 0.031 | −0.128 | −0.007 |
| Ind3 | −0.045 | 0.023 | −0.093 | −0.005 |
| (C1) | −0.202 | 0.063 | −0.323 | −0.078 |
| (C2) | −0.219 | 0.062 | −0.335 | −0.092 |
| (C3) | −0.017 | 0.027 | −0.074 | 0.032 |

Ind1:Job Insecurity→Emotional Exhaustion→Organizational Citizenship Behavior.

Ind2:Job Insecurity→Organizational Identity→Organizational Citizenship Behavior.

Ind3:Job Insecurity→Emotional Exhaustion→Organizational Identification→Organizational Citizenship Behavior.

C1: Ind1 vs Ind2.

C2: Ind1 vs Ind3.

C3: Ind2 vs Ind3.

directly suppresses the demonstration of organizational citizenship behaviors but also indirectly exerts a negative influence through three distinct pathways: emotional exhaustion as a primary mediator, organizational identification as a single mediator, and a chain mediation involving emotional exhaustion→organizational identification. It is particularly significant that emotional exhaustion plays the most prominent mediating role in this influence process, while the chain mediation effect, though small, remains statistically significant. This investigation reveals the complex cognitive-emotional linkage mechanism through which job insecurity affects organizational citizenship behaviors, thereby extending the theoretical scope of research on job insecurity outcomes.

First, the empirical results show that job insecurity significantly and negatively predicts OCB ($\beta = -0.579$, $p < 0.01$). This finding aligns with a robust body of literature [4,38] demonstrating that job insecurity, as a significant occupational stressor, precipitates the erosion of employees' psychological resources, consequently reducing their discretionary resource investments in extra-role behaviors [3]. When employees perceive career instability, they prioritize core resource preservation over making additional organizational investments [39]. This perception can also be interpreted as a violation of the psychological contract, undermining reciprocity norms and leading to reduced discretionary workplace contributions [40,41].

Second, our results highlight the critical mediating roles of both emotional exhaustion and organizational identification. Emotional exhaustion emerged as the strongest mediator (mediating effect = $-0.264$), confirming that job insecurity depletes employees' emotional resources, leading to burnout and subsequent withdrawal from OCB. This is consistent with the stressor-strain framework, where job insecurity acts as a chronic stressor leading to emotional depletion [23,42]. Concurrently, organizational identification also functions as a significant mediator (mediating effect = $-0.062$). Job insecurity threatens an employee's sense of belonging and psychological attachment, weakening their identification with the organization [43]. A diminished organizational identity, in turn, reduces employees' voluntary behavioral contributions as their self-definition is no longer closely tied to the organization's fate [15,44].

Third, and most notably, this study substantiated the chain-mediated mechanism ($\beta = -0.045$), elucidating a complex "emotion-cognition-behavior" pathway. Job insecurity first triggers an affective reaction (emotional exhaustion), which then influences a cognitive assessment (reduced organizational identification), ultimately resulting in a behavioral outcome (decreased OCB). This finding aligns with the core tenets of COR theory, where initial resource loss (from stress) leads to further resource loss and defensive posturing [45]. It demonstrates that emotional states are not merely endpoints but are crucial antecedents to cognitive reappraisals of the employee-organization relationship [46,47]. This dynamic interaction underscores the complexity of job insecurity's impact on employee behavior and provides a more nuanced explanation than single-mediator models.

This study offers several theoretical contributions. Primarily, by testing a chained mediation model, it advances the literature by integrating affective (emotional exhaustion) and cognitive (organizational identification) mechanisms to explain the job insecurity-OCB link. This moves beyond prior research that often examined these mediators in isolation [4,23] and empirically validates a more nuanced "stressor →affective reaction → cognitive reappraisal → behavioral outcome" sequence. Furthermore, this research enriches the application of COR theory and AET. It demonstrates how the threat of resource loss (job insecurity) initiates a loss spiral, starting with emotional resources and extending to cognitive resources (identity), which aligns with COR principles. It also provides clear support for AET by showing how a negative work event (perceived insecurity) triggers emotional responses that subsequently shape work-related attitudes and behaviors [48,49].

From a practical standpoint, our findings provide managers with targeted intervention strategies. The strong mediating effect of emotional exhaustion underscores the urgent need for organizations to implement resource-replenishing initiatives, such as stress management programs, mental health support (e.g., EAPs), and promoting a healthy work-life balance, to buffer the initial impact of job insecurity. Moreover, the significant role of organizational identification highlights the importance of reinforcing the employee-organization bond during uncertain times. Managers can achieve this through transparent communication about organizational changes, clarifying employees' roles and future prospects, and fostering

a culture of belonging. The chained mediation pathway suggests that a dual-pronged approach—simultaneously managing emotional distress and strengthening cognitive attachment—will be most effective in preserving OCB.

Although this study investigated the psychological mechanisms of job insecurity's impact on organizational citizenship behavior using a three-time longitudinal design, the research relies solely on employee self-reported assessments, lacking corroborative evidence from supervisor evaluations or objective behavioral indicators. Additionally, the study was predominantly drawn from enterprises within specific geographical contexts, potentially constraining the findings' external validity. Moreover, the research insufficiently explored the moderating influences of individual dispositional characteristics (e.g., core self-evaluation, psychological resilience) and organizational contextual factors (e.g., organizational support perceptions, leadership styles) on the proposed model. Future research could address these limitations by (1) introducing multi-rater data, such as supervisor-evaluated OCB; (2) expanding the sample to diverse geographical and industrial contexts; (3) developing a moderated mediation framework to explore key boundary conditions. and (4) regarding our control for common method bias (CMB), a limitation exists in our VIF values, which slightly exceeded the stricter 3.3 benchmark. To build upon our findings, future research could incorporate more advanced diagnostics, such as the marker-variable test [50], for a more rigorous assessment.

## 6. Conclusion

This study confirms that job insecurity negatively impacts organizational citizenship behavior, both directly and indirectly. Our findings reveal three indirect pathways: through emotional exhaustion, through organizational identification, and via a chain mediation where job insecurity fosters emotional exhaustion, which in turn diminishes organizational identification, ultimately reducing OCB. Emotional exhaustion was identified as the most dominant mediating mechanism. These results illuminate a complex psychological process driven by an interplay of affective and cognitive reactions to job uncertainty. For organizations, this highlights the critical need to manage employee emotional well-being and strengthen their sense of organizational belonging to mitigate the damaging effects of job insecurity on crucial discretionary behaviors.

## Supporting information

**S1 File. Appendix (Supplementary Data Analysis).**
(DOCX)

**S2 File. Questionnaire.**
(DOCX)

## Acknowledgments

The authors extend their gratitude to the participating organizations and employees who generously contributed their time and insights to this research. Special thanks to the research assistants who aided in data collection and preliminary analysis.

## Author contributions

**Conceptualization:** Jing Zhu.

**Data curation:** Jing Zhu.

**Investigation:** Jing Zhu.

**Software:** Mingfa Yang.

**Writing – original draft:** Jing Zhu.

**Writing – review & editing:** Mingfa Yang.

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
