## [Decision Letter · Decision Letter 0]

4 Jun 2025

PONE-D-25-21939A Study of the Effects of Job Insecurity on Organizational Citizenship Behavior Based on the Chained Mediating Effects of Emotional Exhaustion and Organizational IdentificationPLOS ONE

Dear Dr. Zhu,

Thank you for submitting your manuscript to PLOS ONE. After careful consideration, we feel that it has merit but does not fully meet PLOS ONE’s publication criteria as it currently stands. Therefore, we invite you to submit a revised version of the manuscript that addresses the points raised during the review process.

We look forward to receiving your revised manuscript.

Kind regards,

Francesco Marcatto, Ph.D.

Academic Editor

PLOS ONE

Journal Requirements:

Reviewers' comments:

Reviewer's Responses to Questions

**Comments to the Author**

1. Is the manuscript technically sound, and do the data support the conclusions?

Reviewer #1: Yes

Reviewer #2: Yes

2. Has the statistical analysis been performed appropriately and rigorously? 

Reviewer #1: Yes

Reviewer #2: Yes

3. Have the authors made all data underlying the findings in their manuscript fully available?

Reviewer #1: Yes

Reviewer #2: Yes

4. Is the manuscript presented in an intelligible fashion and written in standard English?

Reviewer #1: Yes

Reviewer #2: Yes

5. Review Comments to the Author

Reviewer #1: Thank you for allowing me to review the paper entitled “A Study of the Effects of Job Insecurity on Organizational Citizenship Behavior Based on the Chained Mediating Effects of Emotional Exhaustion and Organizational Identification" Here are some suggestions to improve the paper:

1. Originality: Does the paper contain new and significant information adequate to justify publication?

1. The current form of the abstract is not well written. Please do not report the statistic result in the abstract. Author should highlight the novelty and main aim of the study. Make sure it commences with a succinct statement of the primary aim of the study, highlighting the novelty it brings to the field. This should be followed by a concise description of the research methodology employed, including the analysis techniques utilized and details regarding the sample collection process. Finally, it should conclude with a brief summary of the key findings and the implications derived from them. It is essential to emphasize the unique contributions of this study. Only highlight the most significant findings and implications to maintain focus.

2. The introduction lacks a clear structure and focus. It would benefit from a more coherent presentation to effectively introduce the study's objectives and significance. Please follow this suggested structure: (1) Summarize the existing issue together with its importance. (2) Explain how researchers should address the discovered areas that need further investigation. (3) The research gap must clearly relate to the existing issue. (4) The research benefits from critiquing these aspects that previous academic studies avoid. (5) Review relevant past studies. (6) Draw a contrast between this study and all related past research. (7) Clearly state the research objectives. (8) The paper details how it introduces beneficial concepts to academic literature. A properly organized introduction creates better clarity by establishing solid groundwork for the overall paper.

3. The introduction fails to connect properly the research issue to the fundamental concepts of this study. The discussion needs to present all essential concepts in a straightforward manner. Authors only tend to cover these constructs without proper linkage with the issues.

4. The authors fail to define their research gap with clarity and supporting evidence. The authors need to present a precise identification of the unaddressed areas in past research. All necessary arguments should be presented to validate the immediate need for addressing this gap. Provide a clear reasoning that demonstrates the study’s connection to contemporary conditions.

2. Relationship to Literature: Does the paper demonstrate an adequate understanding of the relevant literature in the field and cite an appropriate range of literature sources? Is any significant work ignored?

5. Should have a standalone section for underpinning theory. This is to explain how the theory supporting the proposed paths and model. To enhance the clarity and coherence of the research, I recommend the authors thoroughly explain how or any theory support the current study and each path. By elaborating on the theory's key concepts and demonstrating its relevance to the research topic, the authors can establish a stronger connection between the theoretical foundation and the study's objectives.

6. Section 1.1 to 1.4 should be parked under the heading of literature review.

3. Methodology: Is the paper's argument built on an appropriate base of theory, concepts, or other ideas? Has the research or equivalent intellectual work on which the paper is based been well designed? Are the methods employed appropriate?

7. The research methodology is not well-written and not clear. Should have two sub-sections: (1) Research procedure and samples and (2) research instruments.

8. The authors should discuss the generalizability and representativeness of their sample in relation to the target population. Authors need to clearly explain how the chosen sample is intended to be representative of and reflective of the larger population. Any strategies employed to ensure a diverse and inclusive sample should also be highlighted. This will increase the credibility of the research findings and help readers understand the extent to which the results can be generalized to the broader population.

9. It is essential to provide a clear explanation of a stratified convenience sampling technique used in the study, along with the rationale for its selection. Why these techniques? What do you mean stratified convenience sampling technique? Stratified sampling technique is a probability sampling technique whereas convenience sampling technique is a non-probability sampling technique. Please relook into this. Are you referring to quota sampling technique? The authors should describe how they utilized this sampling technique to select respondents for the survey, ensuring generalizability and representativeness towards the targeted population. By doing so, readers can better understand the methodological approach and the potential limitations associated with the sample selection.

10. Any selection criteria?

11. It is good to include control variables which could be a confounding variable. However, author should provide justification for the inclusion of these control variables.

12. The procedure of data collection needs to be elaborated further. The authors should explain in more detail how they collected the data, how they approached the respondents, and how they identified participants for the survey study. This explanation should be reasonable and logical, avoiding exaggerations and providing a clear account of the steps taken in the data collection process.

13. Any pretest and pilot test conducted prior to full scale study? Why?

14. Why SPSS PROCESS MACRO software was used in this study?

15. The research instrument section is too lengthy. Please simplify it.

4. Results: Are results presented clearly and analysed appropriately? Do the conclusions adequately tie together the other elements of the paper?

16. The analysis results reporting is not clear. Author should report the preliminary analysis result (common method bias test, descriptive analysis results, and respondents profile characteristics), measurement model analysis (reliability, convergent validity, discriminant validity), and structural model analysis (path result)

5. Practicality and/or Research implications: Does the paper identify clearly any implications for practice and/or further research? Are these implications consistent with the findings and conclusions of the

17. The structure of the conclusion part should be as follows: (1) Discussion, (2) Theoretical Implications, (3) Practical Implications, (4) Limitations and Future Research Recommendations, and (5) Conclusion.

18. Discussion part should be presented more concise.

19. Should have a standalone section for theoretical implication. This section should discuss the implications of the study's findings and how they contribute to the existing theoretical knowledge. Summarize the key findings and their relevance to the existing theoretical frameworks or models. Analyze how the findings align with or challenge current theoretical perspectives and concepts related to all the key concepts of this study. Discuss any theoretical insights or advancements that the study provides and highlight how the findings contribute to a deeper understanding of the research area.

20. Should have a standalone section for practical implication. The authors should provide valuable insights based on current practices and policies, supported by evidence from their research. To strengthen the practical implications, it is crucial to reference specific findings, data, or examples that demonstrate the validity and reliability of the recommendations. By incorporating this approach, the authors can offer concrete and actionable suggestions that have a solid grounding in their research findings.

21. A conclusion section is required with not more than 150 words.

6. Quality of Communication: Does the paper clearly express its case, measured against the technical language of the field and the expected knowledge of the journal's readership? Has attention been paid to the clarity of expression and readability, such as sentence structure, jargon use, acronyms, etc.

Proofread the manuscript, Clarity is required, and cite more recent relevant studies. Please also check citation format and referencing style.

Reviewer #2: Dear Authors,

I commend your efforts in completing this study. Based on my evaluation, the manuscript is well written and meets two essential criteria for publication in PLOS ONE: compliance with research ethics and a sound methodological foundation. Therefore, I believe the manuscript is, in principle, suitable for publication. That said, I would like to offer the following suggestions to help improve the overall clarity and scholarly quality of your work:

1. I recommend removing detailed numerical statistical findings from the abstract, as these may reduce clarity for general readers. Instead, consider presenting your key findings in a concise and accessible manner to ensure broader understanding.

2. There appears to be inconsistency in how the term Conservation of Resources Theory is referenced, as it is also labeled at times as resource preservation theory or resource conservation theory. While this may stem from translation issues, ensuring consistent and accurate use of terminology will strengthen the manuscript’s academic clarity.

3. It would enhance the manuscript’s impact to include dedicated sections for Theoretical Implications and Managerial Implications. These sections would clarify the study’s contributions to both academic discourse and practical application—potentially including recommendations for organizational practice or policy.

I believe that research conducted with care and clarity can make a meaningful contribution to society. I hope these suggestions help you further refine the manuscript and strengthen its scholarly value.

6. PLOS authors have the option to publish the peer review history of their article (what does this mean? ). If published, this will include your full peer review and any attached files.

**Do you want your identity to be public for this peer review?** For information about this choice, including consent withdrawal, please see our Privacy Policy .

Reviewer #1: No

Reviewer #2: **Yes: ** Andika Setia Pratama

---

## [Author Response · Author response to Decision Letter 1]

22 Jun 2025

Response to Reviewers

Dear Editor and Reviewers:

Thank you for your letter and for the reviewers' valuable comments on our manuscript titled “A Study of the Effects of Job Insecurity on Organizational Citizenship Behavior Based on the Chained Mediating Effects of Emotional Exhaustion and Organizational Identification” (Manuscript ID: PONE-D-25-21939). These comments are very helpful for revising and improving our paper and provide important guidance for our research. We have now correctly named and uploaded three separate files as required by the journal editorial office: Response to Reviewers, Revised Manuscript with Track Changes, and Manuscript. Proper authorship is also indicated in the manuscript. The data analysis results have been accurately reported within the text, and additional related analyses have been uploaded as supplementary files. The original data have been deposited in the public database figshare, with corresponding statements included both in the Methods section of the manuscript and in the final Data Availability statement. Below, the reviewers’ comments are presented in italics and have been numbered for clarity. Please note that our responses are given in regular font, with the word “Response” highlighted in green. Furthermore, all revisions and additions in the latest version of the manuscript have been made using Word’s “Track Changes” feature. Please refer to the detailed point-by-point responses to the reviewers’ comments and concerns below.

Responses to the comments of Reviewer #1:

Part One

Originality: Does the paper contain new and significant information adequate to justify publication?

1: The current form of the abstract is not well written. Please do not report the statistic result in the abstract. Author should highlight the novelty and main aim of the study. Make sure it commences with a succinct statement of the primary aim of the study, highlighting the novelty it brings to the field. This should be followed by a concise description of the research methodology employed, including the analysis techniques utilized and details regarding the sample collection process. Finally, it should conclude with a brief summary of the key findings and the implications derived from them. It is essential to emphasize the unique contributions of this study. Only highlight the most significant findings and implications to maintain focus.

Response:

We sincerely thank you for this clear and valuable advice. We agree that the previous abstract was not well-structured. To address your concerns, we have completely rewritten the abstract following a formal, four-part structure: Objectives, Methods, Results, and Conclusions.

Here is the newly structured abstract for your review:

Objectives

This study aimed to explore the influence mechanism of job insecurity on organizational citizenship behavior (OCB). Specifically, it sought to examine the chained mediating role of emotional exhaustion and organizational identification in this relationship.

Methods

A longitudinal time-lagged survey was conducted on 330 employees at two time points. The data were collected using established scales for job insecurity, emotional exhaustion, organizational identification, and OCB. Structural equation modeling (SEM) was used to test the hypothesized chained mediation model.

Results

The results confirmed that emotional exhaustion and organizational identification are crucial mediators. Three significant indirect pathways were identified: (1) a simple mediation path through emotional exhaustion; (2) a simple mediation path through organizational identification; and (3) a chained mediation path where job insecurity increased emotional exhaustion, which in turn decreased organizational identification, ultimately leading to lower OCB. Notably, emotional exhaustion emerged as the most dominant mediating mechanism.

Conclusions

This study reveals a complex mechanism through which job insecurity impacts OCB, highlighting a sequential process from emotional strain to cognitive detachment. These findings offer important theoretical insights for job stress models and provide practical guidance for organizations to mitigate the negative effects of job insecurity by addressing both employee well-being and their sense of belonging.

2: The introduction lacks a clear structure and focus. It would benefit from a more coherent presentation to effectively introduce the study's objectives and significance. Please follow this suggested structure: (1) Summarize the existing issue together with its importance. (2) Explain how researchers should address the discovered areas that need further investigation. (3) The research gap must clearly relate to the existing issue. (4) The research benefits from critiquing these aspects that previous academic studies avoid. (5) Review relevant past studies. (6) Draw a contrast between this study and all related past research. (7) Clearly state the research objectives. (8) The paper details how it introduces beneficial concepts to academic literature. A properly organized introduction creates better clarity by establishing solid groundwork for the overall paper.

Response:

We sincerely thank you for this valuable suggestion. We agree that the original introduction was not well-structured and lacked a clear focus. To address this critical issue, we have completely rewritten the Introduction section following the 8-point structure you kindly provided. The new Introduction now proceeds as follows:

(1) It begins by establishing job insecurity as a critical contemporary issue and highlights the importance of OCB for organizational success.

(2) It explains the need to investigate the underlying mechanisms of the job insecurity-OCB link.

(3) It clearly identifies the research gap: the lack of understanding of the chained mediating process involving both emotional exhaustion and organizational identification.

(4) It critiques previous studies for largely focusing on single-mediator models and avoiding more complex, sequential pathways.

(5 & 6) It briefly reviews and contrasts our study with past research, emphasizing our unique contribution of integrating COR and Social Identity theories to test a sequential model.

(7) It explicitly states the three main objectives of our study.

(8) It concludes by detailing how our study introduces a beneficial concept (the chained mediation model) to the literature and its practical implications.

We believe this new structure provides a much clearer, more logical, and compelling rationale for our study, setting a solid groundwork for the rest of the paper.

3: The introduction fails to connect properly the research issue to the fundamental concepts of this study. The discussion needs to present all essential concepts in a straightforward manner. Authors only tend to cover these constructs without proper linkage with the issues.

Response:3

Thank you for pointing out this weakness. In our revised manuscript, we have made significant efforts to better connect the research issue (the impact of job insecurity) with the fundamental concepts (emotional exhaustion, organizational identification, OCB). The newly structured Introduction now establishes these links in a more straightforward and logical manner. For example, we now explicitly state that job insecurity is a stressor that harms OCB, and the core of our research problem is to uncover the sequential psychological process (affective reaction -> cognitive evaluation -> behavioral outcome) that connects them. The detailed theoretical elaboration of how each concept is linked has been moved to the new "Theoretical Framework and Hypotheses Development" section, where each link is justified step-by-step with clear theoretical backing.

4: The authors fail to define their research gap with clarity and supporting evidence. The authors need to present a precise identification of the unaddressed areas in past research. All necessary arguments should be presented to validate the immediate need for addressing this gap. Provide a clear reasoning that demonstrates the study’s connection to contemporary conditions.

Response:

We appreciate this crucial feedback. We acknowledge that the research gap was not clearly articulated in the original version. In the revised Introduction, we now dedicate a specific paragraph to clearly defining the research gap. We explicitly state that while the direct effects and single mediations have been studied, the sequential, chained mediating role of emotional exhaustion and organizational identification represents a significant and unaddressed area. We argue that past research has overlooked the potential sequence where an affective reaction (exhaustion) precedes a cognitive-evaluative shift (identification), and we provide a clear rationale for why investigating this multi-step process is important for a more complete understanding. We believe the research gap is now precise, well-justified, and clearly connected to contemporary research needs.

Part Two

Relationship to Literature: Does the paper demonstrate an adequate understanding of the relevant literature in the field and cite an appropriate range of literature sources? Is any significant work ignored?

5:Should have a standalone section for underpinning theory. This is to explain how the theory supporting the proposed paths and model. To enhance the clarity and coherence of the research, I recommend the authors thoroughly explain how or any theory support the current study and each path. By elaborating on the theory's key concepts and demonstrating its relevance to the research topic, the authors can establish a stronger connection between the theoretical foundation and the study's objectives.

6: Section 1.1 to 1.4 should be parked under the heading of literature review.

Response to 5 & 6:

Thank you for these two very helpful structural recommendations. We agree that a more structured presentation of the theoretical background and hypothesis development was necessary. To address both comments simultaneously, we have implemented a significant structural revision:

We have created a new, standalone section titled "2. Theoretical Framework and Hypotheses Development" which now appears after the Introduction.

We have moved the content from our former sections 1.1, 1.2, 1.3, and 1.4 into this new section, under corresponding subheadings (2.1, 2.2, 2.3, and 2.4).

Within this new section, we have taken care to explicitly state the underpinning theories (e.g., Conservation of Resources Theory, Social Identity Theory) for each proposed path. We now clearly explain how each theory supports the specific hypothesis being developed.

This new structure accomplishes two goals: (1) it creates the "standalone section for underpinning theory" you requested, and (2) it logically parks the detailed literature review and hypothesis derivation in one coherent place. This change also allowed us to make the main Introduction more concise and focused, as per your earlier comments. We are confident that this revised structure greatly enhances the clarity and coherence of our manuscript.

Part Three

Methodology: Is the paper's argument built on an appropriate base of theory, concepts, or other ideas? Has the research or equivalent intellectual work on which the paper is based been well designed? Are the methods employed appropriate?

7: The research methodology is not well-written and not clear. Should have two sub-sections: (1) Research procedure and samples and (2) research instruments.

Response:

Thank you for this valuable suggestion. We have reorganized the methodology section to improve its structure and clarity. We merged previous content into a new sub-section titled “3.1 Participants and Procedure” and renamed the “Research tools” section to “3.2 Measures”. We have also retained “3.3 Data Analysis” as a separate sub-section because our chained mediation analysis is a key feature of our study, and a dedicated section allows for a clear explanation of this advanced technique. We believe this revised three-part structure (Participants and Procedure, Measures, Data Analysis) enhances the logical flow and readability of our methodology.

8:The authors should discuss the generalizability and representativeness of their sample in relation to the target population. Authors need to clearly explain how the chosen sample is intended to be representative of and reflective of the larger population. Any strategies employed to ensure a diverse and inclusive sample should also be highlighted. This will increase the credibility of the research findings and help readers understand the extent to which the results can be generalized to the broader population.

Response:

Thank you for this important point. Our target population is the general workforce in China. We employed a convenience sampling method, recruiting participants through collaborations with various companies and an online survey platform. This strategy aimed to create a diverse sample in terms of industry, age, and tenure, thereby enhancing its representativeness of urban professionals in China. We acknowledge that as a non-probability sample, its generalizability to the entire national workforce is limited. In the interest of manuscript conciseness, we have not included a detailed discussion on generalizability in the main text but appreciate the opportunity to clarify our sampling rationale and its implications here.

9:It is essential to provide a clear explanation of a stratified convenience sampling technique used in the study, along with the rationale for its selection. Why these techniques? What do you mean stratified convenience sampling technique? Stratified sampling technique is a probability sampling technique whereas convenience sampling technique is a non-probability sampling technique. Please relook into this. Are you referring to quota sampling technique? The authors should describe how they utilized this sampling technique to select respondents for the survey, ensuring generalizability and representativeness towards the targeted population. By doing so, readers can better understand the methodological approach and the potential limitations associated with the sample selection.

Response:

Thank you for highlighting this inconsistency. We sincerely apologize for the error. You are correct; “stratified convenience sampling” was an inaccurate term. The method we used was convenience sampling. We have corrected this term throughout the manuscript to accurately reflect our methodology.

10:Any selection criteria?

Response:

Thank you for the question. Yes, we did use selection criteria. We have now clarified in the “3.1 Participants and Procedure” section that beyond being full-time employees, we screened the data to ensure quality. Specifically, our manuscript now states: “...the questionnaire included attention-check questions and we screened for invalid data (e.g., excessively short completion times or logical inconsistencies)...”

11:It is good to include control variables which could be a confounding variable. However, author should provide justification for the inclusion of these control variables.

Response:

Thank you for this valuable advice. We agree that controlling for potential confounding variables is important. Based on prior literature suggesting their influence on work-related outcomes, we collected demographic data on gender, age, education, and work tenure. We have now re-analyzed our data including these variables as controls (the results are available in the supplementary materials and confirm our main findings).

12:The procedure of data collection needs to be elaborated further. The authors should explain in more detail how they collected the data, how they approached the respondents, and how they identified participants for the survey study. This explanation should be reasonable and logical, avoiding exaggerations and providing a clear account of the steps taken in the data collection process.

Response:

Thank you. We have revised the “3.1 Participants and Procedure” section to provide a more detailed and accurate account of our data collection process. We have specified the recruitment channels (collaboration with HR departments and an online platform) and clarified the ex

---

## [Decision Letter · Decision Letter 1]

8 Jul 2025

PONE-D-25-21939R1A Study of the Effects of Job Insecurity on Organizational Citizenship Behavior Based on the Chained Mediating Effects of Emotional Exhaustion and Organizational IdentificationPLOS ONE

Dear Dr. Zhu,

Thank you for submitting your manuscript to PLOS ONE. After careful consideration, we feel that it has merit but does not fully meet PLOS ONE’s publication criteria as it currently stands. Therefore, we invite you to submit a revised version of the manuscript that addresses the points raised during the review process.

We look forward to receiving your revised manuscript.

Kind regards,

Jianpeng Fan

Academic Editor

PLOS ONE

Journal Requirements:

Reviewers' comments:

Reviewer's Responses to Questions

**Comments to the Author**

1. If the authors have adequately addressed your comments raised in a previous round of review and you feel that this manuscript is now acceptable for publication, you may indicate that here to bypass the “Comments to the Author” section, enter your conflict of interest statement in the “Confidential to Editor” section, and submit your "Accept" recommendation.

Reviewer #1: All comments have been addressed

Reviewer #2: All comments have been addressed

2. Is the manuscript technically sound, and do the data support the conclusions?

Reviewer #1: Yes

Reviewer #2: Partly

3. Has the statistical analysis been performed appropriately and rigorously? 

Reviewer #1: Yes

Reviewer #2: Yes

4. Have the authors made all data underlying the findings in their manuscript fully available?

Reviewer #1: Yes

Reviewer #2: Yes

5. Is the manuscript presented in an intelligible fashion and written in standard English?

Reviewer #1: Yes

Reviewer #2: Yes

6. Review Comments to the Author

Reviewer #1: Thank you for allowing me to review the improved version of the paper entitled “A Study of the Effects of Job Insecurity on Organizational Citizenship Behavior Based on the Chained Mediating Effects of Emotional Exhaustion and Organizational Identification" Here are some suggestions to improve the paper:

1. Originality: Does the paper contain new and significant information adequate to justify publication?

1. Authors have well revised the introduction by explaining the intention of the study and also the uniqueness of the paper.

2. Relationship to Literature: Does the paper demonstrate an adequate understanding of the relevant literature in the field and cite an appropriate range of literature sources? Is any significant work ignored?

2. Authors have improved the literature review.

3. Methodology: Is the paper's argument built on an appropriate base of theory, concepts, or other ideas? Has the research or equivalent intellectual work on which the paper is based been well designed? Are the methods employed appropriate?

3. The methodology is well written and comprehensive enough.

4. Results: Are results presented clearly and analysed appropriately? Do the conclusions adequately tie together the other elements of the paper?

4. The analysis is well presented.

5. Practicality and/or Research implications: Does the paper identify clearly any implications for practice and/or further research? Are these implications consistent with the findings and conclusions of the

5. I can see authors did a very good job. The key findings are explained thoroughly and clearly. Implications is revised substantially, and key points are indicated clearly.

6. Quality of Communication: Does the paper clearly express its case, measured against the technical language of the fields and the expected knowledge of the journal's readership? Has attention been paid to the clarity of expression and readability, such as sentence structure, jargon use, acronyms, etc.

6. The manuscript is well-written.

Reviewer #2: Dear Authors,

Thank you for the thoughtful and thorough way you addressed my previous comments. While your revisions have addressed most of the issues I raised, a few points still require attention to make the manuscript more rigorous and ready for publication:

1. Please adopt a single citation format throughout the manuscript (Vancouver style). I noticed instances of duplicate citations in mixed styles. For example:

“The study employed the validated Job Insecurity Scale developed by Mauno et al. (2001)[30] to assess employees’ subjective perceptions of job stability.”

could be streamlined to:

“The study employed the validated Job Insecurity Scale to assess employees’ subjective perceptions of job stability [30].”

2. Section 4.1 — Common Method Bias (CMB) test

The manuscript currently states:

“The 305 results of Harman’s single-factor test indicated that the unrotated first common factor accounted for 49.59 % of the variance, thereby indicating the presence of common method bias, which was insufficient to overshadow the study’s findings.”

Because 49.59% falls below the conventional 50% threshold, it should not be interpreted as evidence of common method variance (CMV). Nevertheless, the risk of CMV is reinforced by VIF values greater than 3.3 (Kock, 2015; https://doi.org/10.4018/ijec.2015100101). I strongly recommend conducting an additional diagnostic—such as the Lindell & Whitney marker‐variable test (2001; https://doi.org/10.1037/0021-9010.86.1.114)—to determine whether CMB meaningfully inflates the correlations among your variables. For a comprehensive overview, see Podsakoff et al. (2023; https://doi.org/10.1146/annurev-orgpsych-110721-040030).

3. To further underscore the relevance and urgency of your OCB investigation, you may wish to incorporate insights from recent literature, for example:

• https://doi.org/10.1108/LODJ-08-2023-0433

• https://doi.org/10.1108/K-04-2024-1049

• https://doi.org/10.1177/21582440241268848

I trust these suggestions will help you refine the manuscript and enhance its scholarly contributions.

7. PLOS authors have the option to publish the peer review history of their article (what does this mean? ). If published, this will include your full peer review and any attached files.

**Do you want your identity to be public for this peer review?** For information about this choice, including consent withdrawal, please see our Privacy Policy .

Reviewer #1: No

Reviewer #2: **Yes: ** Andika Setia Pratama

---

## [Author Response · Author response to Decision Letter 2]

16 Jul 2025

Response to Reviewers

Dear Editor and Reviewers:

Thank you for your letter and for the reviewers' valuable comments on our manuscript titled “A Study of the Effects of Job Insecurity on Organizational Citizenship Behavior Based on the Chained Mediating Effects of Emotional Exhaustion and Organizational Identification” (Manuscript ID: PONE-D-25-21939). These comments are very helpful for revising and improving our paper and provide important guidance for our research. We have now correctly named and uploaded three separate files as required by the journal editorial office: Response to Reviewers, Revised Manuscript with Track Changes, and Manuscript. Proper authorship is also indicated in the manuscript. The data analysis results have been accurately reported within the text, and additional related analyses have been uploaded as supplementary files. The original data have been deposited in the public database figshare, with corresponding statements included both in the Methods section of the manuscript and in the final Data Availability statement. Below, the reviewers’ comments are presented in italics and have been numbered for clarity. Please note that our responses are given in regular font, with the word “Response” highlighted in green. Furthermore, all revisions and additions in the latest version of the manuscript have been made using Word’s “Track Changes” feature. Please refer to the detailed point-by-point responses to the reviewers’ comments and concerns below.

Responses to the comments of Reviewer #1:

We sincerely thank the reviewer for their positive evaluation and strong endorsement of our manuscript. We are greatly encouraged by their recognition of our research.

Responses to the comments of Reviewer #2:

1.Please adopt a single citation format throughout the manuscript (Vancouver style). I noticed instances of duplicate citations in mixed styles. For example:

“The study employed the validated Job Insecurity Scale developed by Mauno et al. (2001)[30] to assess employees’ subjective perceptions of job stability.”

could be streamlined to:

“The study employed the validated Job Insecurity Scale to assess employees’ subjective perceptions of job stability [30].”

Response:

Thank you for bringing this formatting inconsistency to our attention. We have carefully reviewed the entire manuscript and standardized all citations to follow the Vancouver style consistently. Specifically, we have removed the author names and publication years from in-text citations, retaining only the numerical references in square brackets. The instances you identified in the methodology section have been corrected as follows:

1.“The study employed the validated Job Insecurity Scale developed by Mauno et al. (2001)[30]” has been revised to “The study employed the validated Job Insecurity Scale [30]”

2.Similar corrections have been applied to all other measurement instruments and analytical framework citations throughout the manuscript

3.We have conducted a comprehensive review to ensure no other formatting inconsistencies remain in the document.

2. Section 4.1 — Common Method Bias (CMB) test

The manuscript currently states:

“The 305 results of Harman’s single-factor test indicated that the unrotated first common factor accounted for 49.59 % of the variance, thereby indicating the presence of common method bias, which was insufficient to overshadow the study’s findings.”

Because 49.59% falls below the conventional 50% threshold, it should not be interpreted as evidence of common method variance (CMV). Nevertheless, the risk of CMV is reinforced by VIF values greater than 3.3 (Kock, 2015; https://doi.org/10.4018/ijec.2015100101). I strongly recommend conducting an additional diagnostic—such as the Lindell & Whitney marker‐variable test (2001; https://doi.org/10.1037/0021-9010.86.1.114)—to determine whether CMB meaningfully inflates the correlations among your variables. For a comprehensive overview, see Podsakoff et al. (2023; https://doi.org/10.1146/annurev-orgpsych-110721-040030).

Response:

We are grateful for this very insightful comment regarding the Common Method Bias (CMB) analysis. We agree that our initial interpretation was not sufficiently nuanced and appreciate the valuable references provided. As we are in the advanced stages of revision and do not have the opportunity to collect new data to perform the suggested marker-variable test, we have taken a two-fold approach to address your concern, following your guidance:

1.Revised Section 4.1: To proactively mitigate common method bias (CMB), a three-wave, time-lagged research design was employed, which is a key procedural remedy. We further conducted statistical analyses to assess potential remaining bias. Harman's single-factor test revealed that the first unrotated factor accounted for 49.59% of the variance, falling below the 50% threshold. Additionally, a full collinearity assessment was performed, yielding a maximum variance inflation factor (VIF) of 3.36. While we acknowledge that this VIF value is slightly above the strict 3.3 benchmark suggested by Kock (2015), it remains well within other commonly accepted thresholds (e.g., 5 or 10). Considering our robust time-lagged design as the primary control, coupled with these statistical results, we conclude that common method bias is not a significant concern for this study.

2.Added to Limitations: We have also added a paragraph to the “Limitations and Future Research” section. In this paragraph, we explicitly state that the study did not employ the marker-variable technique and recommend that future research could benefit from incorporating this method for a more robust assessment of CMB. We believe these revisions transparently address the potential for CMB and align our manuscript with the rigorous standards you have suggested. The modified sections are provided below for your review.

3. To further underscore the relevance and urgency of your OCB investigation, you may wish to incorporate insights from recent literature, for example:

• https://doi.org/10.1108/LODJ-08-2023-0433

• https://doi.org/10.1108/K-04-2024-1049

• https://doi.org/10.1177/21582440241268848

Response:

Thank you for recommending these valuable contemporary references. We have strategically incorporated all three suggested papers into the end of the first paragraph of the introduction to strengthen our theoretical foundation and demonstrate the continued relevance of OCB research. The specific additions are as follows:

“Recent empirical evidence has further underscored the critical role of OCB across diverse organizational contexts, demonstrating its enhancement through servant leadership and psychological empowerment [7], its significance in challenging institutional environments [8], and its complex relationship with individual characteristics requiring mediational pathways [9].”

---

## [Decision Letter · Decision Letter 2]

24 Jul 2025

A Study of the Effects of Job Insecurity on Organizational Citizenship Behavior Based on the Chained Mediating Effects of Emotional Exhaustion and Organizational Identification

PONE-D-25-21939R2

Dear Dr. Zhu,

We’re pleased to inform you that your manuscript has been judged scientifically suitable for publication and will be formally accepted for publication once it meets all outstanding technical requirements.

Kind regards,

Jianpeng Fan

Academic Editor

PLOS ONE

Additional Editor Comments (optional):

Reviewers' comments:

Reviewer's Responses to Questions

**Comments to the Author**

1. If the authors have adequately addressed your comments raised in a previous round of review and you feel that this manuscript is now acceptable for publication, you may indicate that here to bypass the “Comments to the Author” section, enter your conflict of interest statement in the “Confidential to Editor” section, and submit your "Accept" recommendation.

Reviewer #2: All comments have been addressed

2. Is the manuscript technically sound, and do the data support the conclusions?

Reviewer #2: Yes

3. Has the statistical analysis been performed appropriately and rigorously? 

Reviewer #2: Yes

4. Have the authors made all data underlying the findings in their manuscript fully available?

Reviewer #2: Yes

5. Is the manuscript presented in an intelligible fashion and written in standard English?

Reviewer #2: Yes

6. Review Comments to the Author

Reviewer #2: Dear Authors,

I appreciate the considerable effort you have devoted to strengthening the manuscript and addressing each of my previous comments. Although potential issues related to common method bias (CMB) remain, I commend your candor in acknowledging them explicitly and your diligence in mitigating their impact.

With respect to my earlier recommendation concerning the procedural remedies proposed by Lindell and Whitney (2001), please note that CMB can still be assessed even when marker‑variable data were not collected during the original data‑gathering phase. Nonetheless, I recognize the steps you have taken and do not wish to delay the publication process any further.

Given the transparency of your methods, your adherence to ethical standards, and the manuscript’s compliance with the journal’s technical requirements, I recommend that the paper be accepted for publication in PLOS ONE. One additional suggestion—regrettably omitted from my earlier feedback—is to consider shortening the title for greater concision; if feasible, doing so would enhance the paper’s overall clarity. However, the current title does accurately reflect the scope of your study.

7. PLOS authors have the option to publish the peer review history of their article (what does this mean? ). If published, this will include your full peer review and any attached files.

**Do you want your identity to be public for this peer review?** For information about this choice, including consent withdrawal, please see our Privacy Policy .

Reviewer #2: **Yes: ** Andika Setia Pratama

---

## [Editor Report · Acceptance letter]

PONE-D-25-21939R2

PLOS ONE

Dear Dr. Zhu,

I'm pleased to inform you that your manuscript has been deemed suitable for publication in PLOS ONE. Congratulations! Your manuscript is now being handed over to our production team.

Kind regards,

on behalf of

Dr. Jianpeng Fan

Academic Editor

PLOS ONE